# Enhanced atmospheric oxidation toward carbon neutrality reduces methane's climate forcing

Mingxu Liu [1,2], Yu Song [1], Hitoshi Matsui [2] ✉, Fang Shang[1], Ling Kang[1], Xuhui Cai[1], Hongsheng Zhang [3] & Tong Zhu [1] ✉

The hydroxyl radical (OH), as the central atmospheric oxidant, controls the removal rates of methane, a powerful greenhouse gas. It is being suggested that OH levels would decrease with reductions of nitrogen oxides and ozone levels by climate polices, but this remains unsettled. Here, we show that driven by the carbon neutrality pledge, the global-mean OH concentration, derived from multiple chemistry-climate model simulations, is projected to be significantly increasing with a trend of 0.071–0.16% per year during 2015–2100. The leading cause of this OH enhancement is dramatic decreases in carbon monoxide and methane concentrations, which together reduce OH sinks. The OH increase shortens methane's lifetime by 0.19–1.1 years across models and subsequently diminishes methane's radiative forcing. If following a largely unmitigated scenario, the global OH exhibits a significant decrease that would exacerbate methane's radiative forcing. Thus, we highlight that targeted emission abatement strategies for sustained oxidation capacity can benefit climate change mitigation in the Anthropocene.

The atmospheric hydroxyl radical (OH) originates primarily from the collision of excited oxygen atom (O($^1$D)) produced by ozone (O$_3$) photolysis with water vapor, and secondarily from radical recycling in the presence of nitrogen oxides (NO$_x$)[1,2]. As the most important cleaning agent and oxidant in the troposphere, OH can immediately react with various trace gases, such as methane (CH$_4$), volatile organic compounds (VOCs), and carbon monoxide (CO)[3–5], and accordingly regulate their lifetimes and climate effects[6–8]. The spatiotemporal evolution of OH therefore plays a critical role in climate projections. Yet, it remains ambiguous how global OH will evolve with changing human activities in the Anthropocene[9–12]. A range of competing factors that determine OH sources and sinks (referred to as loss of OH via reaction with those trace gases) complicate the prediction of OH levels in the global troposphere[9,12–14].

Future evolution of OH levels is probably linked to implementation of climate policies. The Paris Agreement calls for international

cooperation to limit the global temperature rise below 1.5–2.0 degrees relative to preindustrial levels[15,16]. Driven by the carbon neutrality pledges, the socio-economic development is likely transited to a clean energy-based society by the second half of 21st Century, with a phase-out of fossil fuels[17–19]. The targeted measures for carbon neutrality can meanwhile result in substantial reductions of near-term climate forcers (NTCFs) resulting from fossil fuel combustion, including aerosols, CO, NO$_x$, and VOCs[20–22]. The projected reductions in NO$_x$ and reactive carbon species is bound to weaken O$_3$ formation on a global scale, because NO$_2$ photolysis is a major source of O$_3$ in the troposphere[23]. One could therefore expect that atmospheric OH levels will decline accordingly following a reduction in O$_3$ and NO$_x$[24–26]. However, the simultaneous decreases in CH$_4$, CO, and VOCs emissions likely in part or fully balance the OH budget due to a reduction in OH sinks[27,28], such that its future concentrations potentially remain at present day levels. In addition, the potential changes in physical climate including

[1]State Key Joint Laboratory of Environmental Simulation and Pollution Control, College of Environmental Sciences and Engineering, Peking University, Beijing 100871, China. [2]Graduate School of Environmental Studies, Nagoya University, Nagoya, Japan. [3]Laboratory for Atmosphere-Ocean Studies, Department of Atmospheric and Oceanic Science, School of Physics, Peking University, Beijing 100871, China. ✉e-mail: matsui@nagoya-u.jp; tzhu@pku.edu.cn

temperature and specific humidity would affect OH through impacts on its sources and sinks[29,30].

Here, we analyze multiple model simulation results from Coupled Model Intercomparison Project Phase 6 (CMIP6)[31,32] to reveal the trends in global OH concentration following divergent climate scenarios in the 21st century and to identify associated anthropogenic and natural drivers. Then, by reconstructing the decadal variability in global $CH_4$ concentrations induced by OH evolution, we estimate $CH_4$ radiative forcing linked to OH and draw implications for the policy-making of climate change mitigation. Our results shed light on the critical role of the future course in OH, resulting from changing human activities and physical climate driven by the carbon neutrality pledge, in shaping the $CH_4$ trends and its climate forcing.

## Results

### Global OH evolution toward carbon neutrality

Our analysis begins with the model projections of the interannual variability of the global mean OH concentration and its link with key species related to OH sources and sinks within the troposphere, based on three ocean-atmosphere coupled Earth system models that participated in CMIP6, i.e., UKESM[33], GFDL-ESM[34], and MRI-ESM[35] (see Methods). The model results are focused on the period of 2015–2100 in a strong climate mitigation scenario towards sustainable development, Shared Socio-economic Pathway 1–26 (SSP126)[36]. As a comparison, we also analyze the results in the SSP370 scenario, which represents a high-emissions scenario with continued increases in $CO_2$ and $CH_4$.

The SSP126 scenario, used here as a proxy for the carbon neutrality world, entails substantial mitigation of $CH_4$ and other

NTCFs from anthropogenic sources involving fossil fuel extraction, agricultural production, and landfills sectors[20,37]. The reduction in the global mean $CH_4$ concentration according to the SSP126 scenario amounts to approximately 40% in total from 2015 to 2100 (Supplementary Fig. 1). Due to the rapid reduction in $NO_x$ and CO emissions, the models reveal strong decreases in the tropospheric $NO_x$ and CO concentrations across models (Fig. 1). These decreases in $O_3$ precursors jointly weaken the formation of tropospheric $O_3$, which exhibits a significantly downward trend of 0.17–0.28% $yr^{-1}$ ($P < 0.001$) relative to the 2015 levels (Fig. 1c). The zonal mean $O_3$ decreases are widespread throughout the troposphere among three models (Supplementary Fig. 2). Interestingly, while tropospheric $O_3$ concentrations decrease in the SSP126 pathway, the global OH concentrations are significantly increased, with an inter-annual trend of 0.071–0.16% $yr^{-1}$ ($P < 0.001$) over 2015–2100 (Fig. 1d). Even though the simulated global OH ($O_3$) burdens differ among models, they all exhibit an upward (downward) trend during the given period (Supplementary Fig. 3). By contrast, the global OH abundance in the SSP370 scenario exhibits a decreasing trend of 0.037–0.15% $yr^{-1}$ ($P < 0.001$) over 2015–2100 among these models, together with an increase in tropospheric $O_3$ of 0.11–0.17% $yr^{-1}$ ($P < 0.001$) (Supplementary Figs. 3 and 4). This decrease in OH is associated with the doubling of the $CH_4$ concentration that strongly consumes OH (Supplementary Fig. 1). Overall, global OH increases with the concurrent reductions in $NO_x$ and reactive carbon species during the pathway toward carbon neutrality, in contrast to the decreasing trend in an unmitigated scenario.

The projected increases in global OH under the carbon neutrality scenario (SSP126) are rooted in a range of competing factors that control the sources and sinks of OH[2,5]. Our results show an overall

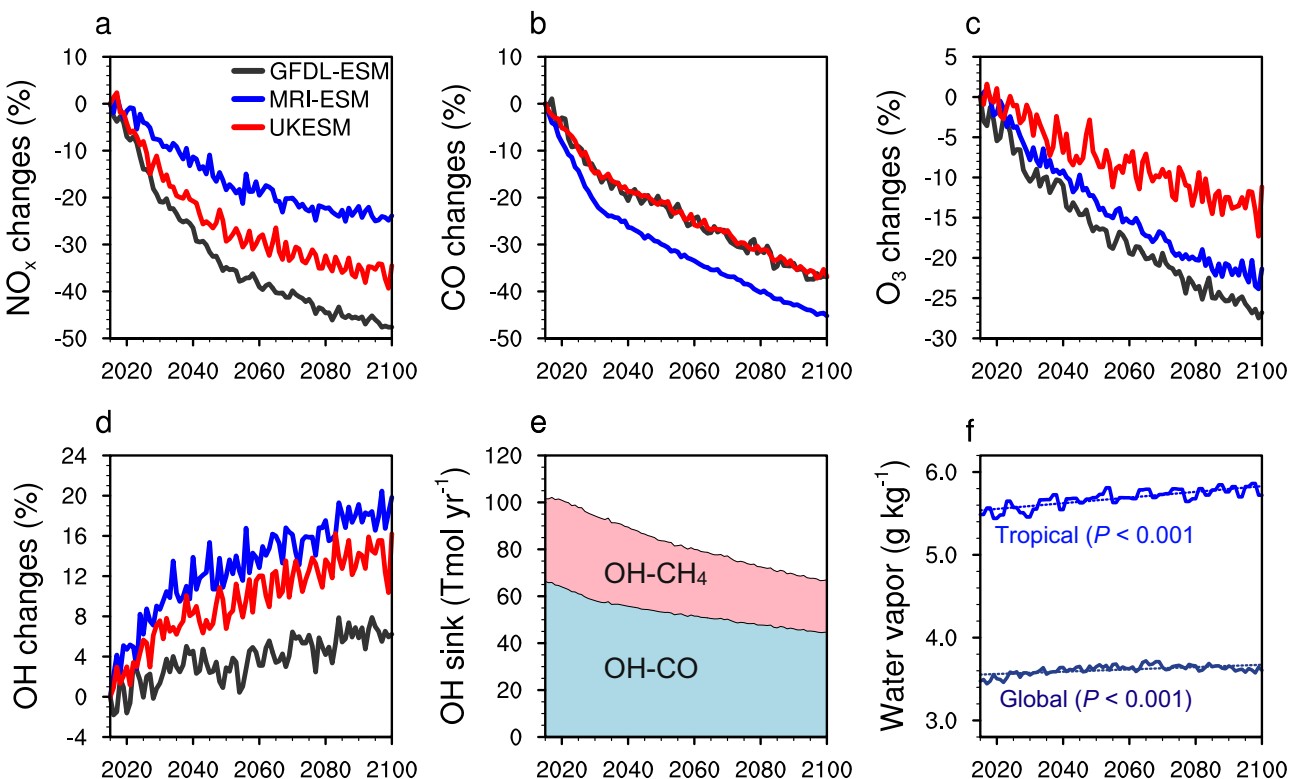

**Fig. 1 | Inter-annual variability in global tropospheric hydroxyl (OH) radical and other reactive trace gases for the period of 2015–2100.** The panels illustrate the percentage changes in (**a**) nitrogen oxides ($NO_x$), (**b**) carbon monoxide (CO), (**c**) ozone ($O_3$), and (**d**) OH burdens compared to the corresponding 2015 levels; (**e**) Trends in OH sinks with respect to CO and $CH_4$ from 2015 to 2100 derived from the ensemble mean model results following the Shared Socioeconomic Pathway 1–2.6 (SSP126) scenario; and (**f**) Trends in global and tropical (20°S–20°N) tropospheric

mean water vapor concentrations, with the significance ($P$) shown in the right panel. The results are provided by the climate projections from three Earth system models (i.e., UKESM, MRI-ESM, and GFDL-ESM) following the SSP126 scenario, except the water vapor mixing ratios that are available only from MRI-ESM. The model diversity is illustrated by the colored lines (legend in the top left panel). We use the Mann-Kendall non-parametric test to reveal the significance of the trends for each species in the main text.

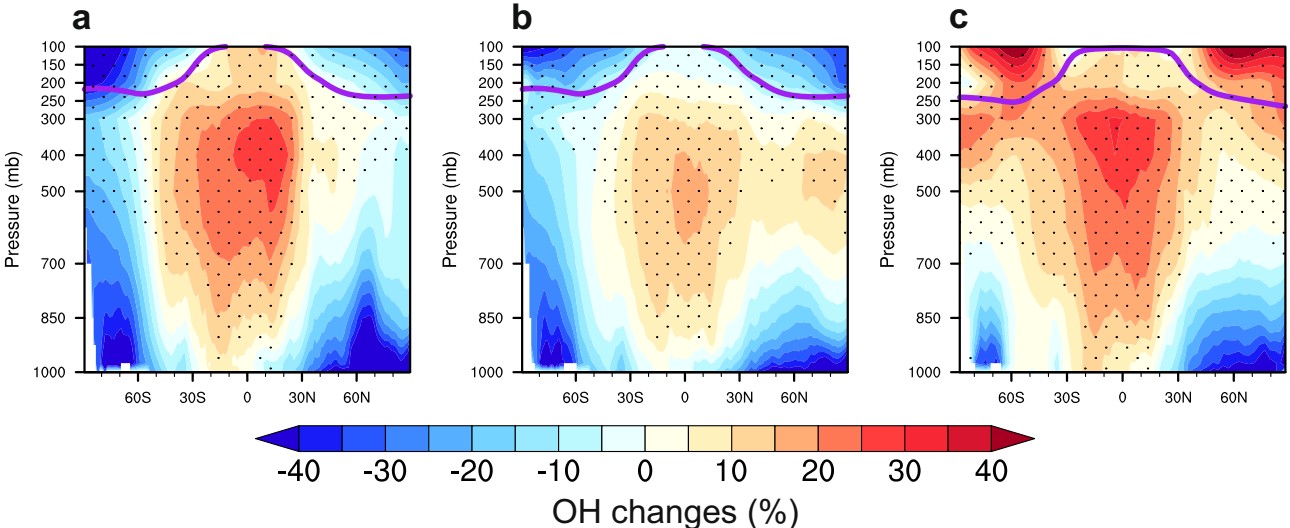

**Fig. 2 | Zonal mean changes in hydroxyl radical (OH) concentrations from 2015 to 2100.** The percentage changes of annual-mean OH from the three model results (**a** UKESM; **b** GFDL-ESM; **c** MRI-ESM) are presented in the zonal mean under the Shared Socioeconomic Pathway 1–2.6 scenario. The overlaying dot pattern indicates positive shift in the ratio of $NO_x$ to CO mixing ratios, used as a proxy of relative changes in OH sources and sinks. The purple solid lines denote the tropopause.

decline in the tropospheric annual-mean $O_3$ of 11–27% across models in the given period, with a multi-model average of 19%. Based the diagnosis of $O_3$ photolysis to form excited oxygen atom in CMIP6 models (see Methods), we calculate that the OH primary production is decreased by 16–26 Tmol $yr^{-1}$ with the reduction of $O_3$ photolysis, equivalent to 16–22% of the 2015 levels. Note that tropospheric water vapor, through its collision with excited oxygen atom, could perturb the primary production of OH in a warmer climate[29,38]. The simulation results available in MRI-ESM show increases of tropospheric mean water vapor content by 3.8% globally and by 5.2% in the tropics (20°S–20°N) from 2015 to 2100 in the SSP126 scenario (Fig. 1f), as a result of enhanced surface evaporation with increased temperature[2]. Given such water vapour increase, the OH production is estimated to be increased by 3.4–4.3 Tmol $yr^{-1}$ across models. Additionally, the OH recycling in the presence of $NO_x$, as another key driver of OH (Eq. 1), is also diminished due to the pronounced decrease in $NO_x$ concentration (Fig. 1a). It is estimated that the tropospheric production flux of OH through the reaction between NO and hydroperoxy radical ($HO_2$) is decreased by 5.8–20 Tmol $yr^{-1}$. The total decrease in annual OH production dominated by $O_3$ photolysis and radical recycling by $NO_x$ is 28–47 Tmol $yr^{-1}$ from 2015 to 2100 (Supplementary Fig. 5).

$$HO_2 + NO \rightarrow NO_2 + OH \qquad (1)$$

On the other hand, we find that the total OH sink with respect to CO and $CH_4$ exhibits a steadily decreasing trend (Fig. 1e), in which OH loss via the reaction with CO is greater. Under the SSP126 scenario, the reduction in $CH_4$ emissions can enhance OH concentrations, shorten $CH_4$ lifetimes, and amplify $CH_4$ decreases, which is known as the $CH_4$ self-feedback[39]. Because both $CH_4$ and its oxidation products (including CO) can consume OH, the $CH_4$ control contributes largely to the OH increase. Apart from these diagnosed outputs from the CMIP6 models, our estimate based on the prognostic variables including the mixing ratios of CO, $CH_4$, formaldehyde, and $O_3$ shows a total decrease in OH sinks of 59–64 Tmol $yr^{-1}$ from 2015 to 2100 (Supplementary Fig. 5). In conclusion, the reduction of OH sink consistently outweighs that of OH sources in each model projection, which accounts for the upward tendency of global OH. These results also demonstrate that the reduction of chemical loss of OH with respect to CO and $CH_4$ is the leading cause of the global OH growth toward the carbon neutrality

scenario, while the increase in tropospheric water vapor content may be also important.

The models indicate a strong dependence of the sign and magnitude of OH changes on both latitude and altitude (Fig. 2), albeit with the overall increase in the global mean OH during the period of decarbonization. As depicted in the zonal mean distributions of OH differences in percentage, the increases in OH mixing ratios under the SSP126 scenario mainly occur in the tropical and subtropical regions, with local increases up to 30% in the free troposphere. However, the OH mixing ratio features general decreases within temperate and polar zones (> 45 degrees), where it drops over 40% locally in the lower to middle troposphere. Because $NO_x$ and CO regulate the production and loss of OH on a global scale, respectively, we employ the ratio of $NO_x$ to CO mixing ratios, an indicator suggested in existing studies[8,40], to infer the drivers of OH tendencies (marked with dots in Fig. 2). We find that increases in this ratio occur mainly in the tropics, in general coinciding with the increase in OH concentrations. As both $NO_x$ and CO widely decline in the SSP126 scenario, the positive shifts in the $NO_x$-to-CO ratio suggest faster decreases in CO concentrations and subsequently in OH sinks with respect to it, resulting in a net increase in OH. With regards to the large decrease of OH near the surface at the middle and high latitudes, the reduction of OH recycling via $NO_x$ (R1) in response to the $NO_x$ reduction and the pronounced $O_3$ decreases (Supplementary Fig. 2) are the primary drivers.

## OH-mediated $CH_4$ radiative forcing

One of the most important implications of increasing OH on the path to carbon neutrality is its potential impact on $CH_4$'s climate effect through changes in $CH_4$ concentrations. We first diagnose the whole-atmosphere chemical lifetime of $CH_4$ by dividing the total $CH_4$ burden by the integrated loss flux over the whole model domain by referring to previous studies[9,29]. The trends in total $CH_4$ lifetime are then estimated by additionally including the lifetime with respect to soil uptake of 150 years[41]. It can be seen that the lifetime trends are closely linked to those divergent OH trends (Fig. 3a). Specifically, with the increase in OH following the SSP126 scenario, the chemical loss of $CH_4$ is enhanced and consequently its lifetime (a multi-model mean plus spread) is shortened from 8.5 (7.4–9.9) year in 2015 to 7.8 (6.6–8.8) year in 2100. Spatially, the variation of $CH_4$ loss rates also tracks that of OH concentrations in the global troposphere (Supplementary Fig. 6).

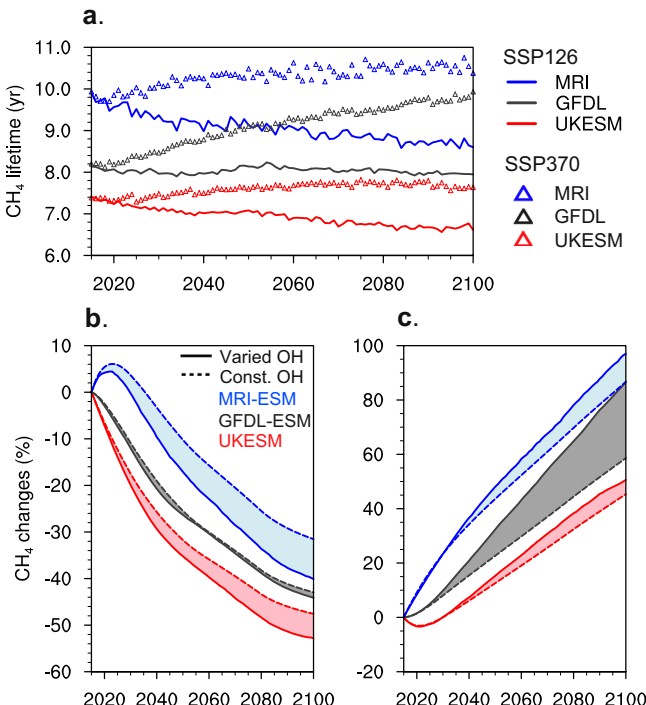

**Fig. 3 | Trends in methane (CH₄) lifetime and concentration from 2015 to 2100.**
**a** The diagnosed CH₄ lifetimes in the SSP126 and SSP370 scenarios based on three models' results. **b** The evolution of the global CH₄ concentrations reconstructed separately with varied hydroxyl radical (OH) (solid lines) and constant OH (dashed lines) for the Shared Socioeconomic Pathway 1–2.6 scenario; (**c**) the same with (**b**) but for the Shared Socioeconomic Pathway 3–7.0 scenario. The effect of temperature changes is also included in our calculation. The comparison of modeled CH₄ trends with the CMIP6 prescribed data is shown in Supplementary Fig. 7.

In line with the spatial patterns of OH changes, the CH₄ chemical loss rates are enhanced locally by over 20% in the tropics, while reduced as much as 40% over the high latitudes.

Along with the OH changes, atmospheric temperature increases, i.e., global warming, can also shorten the CH₄ lifetime by boosting the reaction rate of CH₄ with OH. In our examined case, i.e., SSP126 (low forcing scenario), the global-mean surface temperature increases from 2015 to 2100 are 0.53 K–1.6 K across models. Using the grid-resolved temperature changes from 2015 to 2100 and the CH₄ and OH concentrations from the CMIP6 outputs (see Methods), we estimate that the CH₄ lifetimes are shortened by 0.15–0.27 years due to temperature increases alone. The corresponding estimates due to OH are 0.53–1.4 years. The OH increase is the dominant driver for reductions in CH₄ lifetimes in this scenario, while the temperature increase is also important.

In contrast, the CH₄ lifetime is prolonged by 0.43–1.7 years in total in the SSP370 scenario, due primarily to the decline in global OH. Compared to the SSP126 case, the larger temperature and water vapor increases under accelerated global warming partly masks the longer lifetime induced by the CH₄ self-feedback. We estimate that in this high forcing scenario, the temperature increases shorten the CH₄ lifetimes by 0.74–1.1 years from 2015 to 2100, and the OH primary production is enhanced by about 25% due to water vapor increases. It has been suggested that CH₄ increases induce a negative climate feedback on itself via global warming[29,30].

We next identify the degree to which the CH₄ lifetime changes dominated by OH increases can shape the CH₄ trends in the SSP126 scenario. The global mean CH₄ concentrations are reconstructed with the inter-annual emission data and varying lifetimes using a theoretical box model (see Methods). The lifetime effects can

then be isolated by comparing the CH₄ estimate using varying CH₄ lifetime with that using constant lifetime (fixed at 2015). Dependent on the inter-model diversity in CH₄ lifetimes, the changes in CH₄ concentrations calculated using constant lifetime range from −31% to −43% between 2015 and 2100 (dashed lines in Fig. 3b). The reductions are further amplified by introducing the effects of increased OH and temperature, which together shorten the CH₄ lifetime and lower CH₄ concentrations in this scenario. We estimate that CH₄ mixing ratios are further reduced by 22 ppb to 159 ppb with decreasing lifetimes (solid lines in Fig. 3b), equivalent to 2.1% to 13% relative to the 2015 levels. In contrast, the prolonged lifetime of CH₄ in the SSP370 scenario translates into an increase of CH₄ concentrations by 3.6% to 18% relative to the 2015 levels (Fig. 3c), in addition to those changes induced directly from the increased CH₄ emissions. These results demonstrate that the OH evolution and its impacts on CH₄ lifetime can markedly alter the global CH₄ concentration by the year 2100.

Note that the CMIP6 prescribed CH₄ concentrations were derived from the reduced-complexity climate simulations, which roughly consider the OH evolution and resulting CH₄ lifetime changes[42]. That predicted CH₄ decrease by 2100 for the SSP126 scenario is generally lower than our estimates of CH₄ concentrations using the same CH₄ emissions but with diagnostic lifetimes from the complex climate models (Supplementary Fig. 7). As detailed atmospheric chemical processes are lacked in the reduced-complexity climate model, the derived CH₄ trends used for CMIP6 projections may need to be revisited. Moreover, the concentration-driven mode to simulate CH₄ in CMIP6 models cannot reflect the dynamic responses of CH₄ to tropospheric chemistry in their own experiments. Since those models simulate a wide range of OH values and corresponding CH₄ lifetimes (Fig. 3a), the inferred CH₄ emissions will differ in the future. We show that the declining trends in CH₄ burdens are different among models; those with higher OH levels (shorter CH₄ lifetimes) have larger percent decreases in CH₄ concentrations during 2015–2100 (Fig. 3b). The more realistic changes in CH₄ can be obtained from the emission-driven simulation for CH₄, which enables a full coupling of CH₄-CO-OH in the atmosphere and allows CH₄ concentrations freely evolves[43]. Providing the decreasing CH₄ emissions, the OH enhancement and resulting CH₄ lifetime changes in the emission-driven simulations will be greater than presented in the prescribed data. The next phases of CMIP are encouraged to carry out the inter-model comparison of CH₄ simulations using the emission-driven mode.

Radiative forcing (RF) is a critical index that measures the change in the Earth radiative budget due to an imposed perturbation, such as the emission of anthropogenic greenhouse gases and other NTCFs, and can represent the contributions of different drivers on climate warming[44]. We calculate the temporal variations of global-mean CH₄ RF in the 21st century using the reconstructed global CH₄ (see Methods) and then evaluate the associated OH effects by comparing the case with varied lifetime to that with constant lifetime (fixed at 2015). Our calculation of the present-day (2015) CH₄ RF is 0.62 W m⁻², lying in the confidence range given in the IPCC AR6 report[45]. Adding the OH effects (along with temperature changes) on CH₄ evolution results in more negative changes in CH₄ RF by 2100 for the SSP126 scenario and more positive changes for the SSP370 (Fig. 4a). Specifically, due to the reductions in CH₄ emissions in the SSP126 scenario, the CH₄ RF in 2100 is estimated to be 0.15 to 0.34 W m⁻² with constant OH, but decreases further to 0.089 to 0.26 W m⁻² with increased OH and shortened CH₄ lifetime. Thus, the net effects on CH₄ RF are −0.013 to −0.084 W m⁻² in 2100 among models, equivalent to 6.2% to 41% of the absolute CH₄ RF when excluding the lifetime change, while they turn to be +0.034 to +0.16 W m⁻² for the SSP370 (Fig. 4b). These results suggest that the increase in the OH level on the path to carbon neutrality results in an appreciable cooling effect on climate via enhanced CH₄ removal; however, under a heavy emission scenario, the CH₄ forcing is amplified by decreased OH.

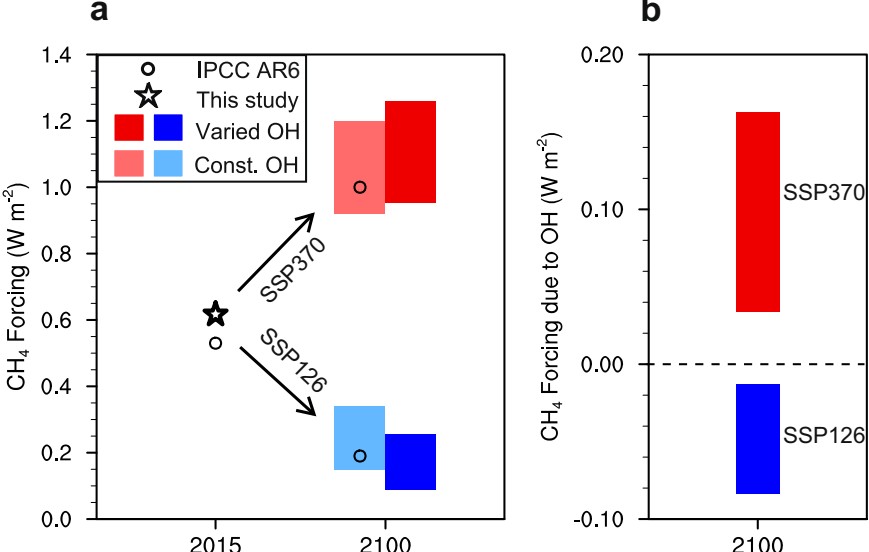

**a**  **b**

Fig. 4 | **CH$_4$ radiative forcing mediated by hydroxyl radical (OH) variations from 2015 to 2100. a** CH$_4$ forcing at 2015 and 2100 following the Shared Socioeconomic Pathway 1–2.6 (SSP126) scenario (blue and light blue bars represent the intermodal diversity of CH$_4$ forcing estimated using varied OH and constant OH, respectively) and the Shared Socioeconomic Pathway 3–7.0 (SSP370) scenario (red and light red bars). **b** The contributions of OH variations on CH$_4$ forcing from 2015 to 2100. The effect of temperature changes is combined with the OH effect on CH$_4$ in our calculation. The positive value means the decrease in CH$_4$ forcing by the OH change. The asterisk marks our estimate of CH$_4$ forcing at 2015 and the black circles mark the values of CH$_4$ effective radiative forcing reported by the Sixth Assessment Report of the Intergovernmental Panel on Climate Change (IPCC AR6).

## Discussion

Our findings suggest that the expectation that considerable NO$_x$ reductions by climate polices would make OH lower in the coming decades may not be true. We identify a range of competing factors that alter OH and demonstrate that the strong decreases in global CO and CH$_4$ during a pathway to carbon neutrality can diminish OH sinks and consequently result in a net increase in OH concentration from 2015 to 2100, despite a significant decrease in O$_3$. This nonlinear coupling of CO-O$_3$-OH-CH$_4$ driven by changing human activities regulates OH atmospheric abundance throughout the 21st century. The changes in atmospheric temperature and water vapour concentrations under a warmer climate also participate in this chemical coupling. Such future trajectory towards carbon neutrality has the added benefit for CH$_4$ mitigation.

Though the world pledges to reach net-zero carbon emissions in the middle of the century, it is still a challenge to deliver on this pledge; due to this, anthropogenic emissions might not perfectly follow the predefined climate scenarios. Their long-term trends will depend on the degree to which targeted climate polices as well as air pollution control measures can be implemented both locally and globally. Specifically, CH$_4$ emission regulation remains a tricky problem. The recent surge in atmospheric CH$_4$ growth rates emphasizes the importance of reinforcing control on CH$_4$ from anthropogenic sources. CH$_4$ is emitted not only from oil and gas industry, but also from agricultural production and landfills. In the unmitigated scenario, SSP370, the agricultural sector will become the dominant emitter of CH$_4$ in the second half of the 21st century. We find that the rapidly increased CH$_4$ concentration in this case would induce a decrease in OH, which then accelerates the rise of CH$_4$ due to the positive CH$_4$ self-feedback on its lifetime and amplifies CH$_4$ radiative forcing by up to 0.16 W m$^{-2}$.

Our study highlights the importance of accounting for the changes in the atmospheric oxidizing capacity when formulating climate policies to control greenhouse gas emissions, as the concomitant reductions of short-lived trace species, including NO$_x$ and reactive carbon species, can collectively determine OH concentrations and thus impact CH$_4$ budgets. A sustained OH level, achieved by a delicate balance between OH sources and sinks, is beneficial for climate change mitigation in the pathway toward a carbon neutral world.

## Methods
### CMIP6 projections

Three state-of-the-art Earth system models that participated in the AerChemMIP CMIP6 experiments[46] are analyzed for OH tendencies: Geophysical Fluid Dynamics Laboratory Earth System Model version 4 (GFDL-ESM)[34], United Kingdom Earth System Model version 1.0 with low resolution (UKESM)[33], and the Meteorological Research Institute Earth System Model Version 2 (MRI-ESM)[35] (Supplementary Table 1). These Earth system models with coupled atmosphere-ocean configuration are capable of representing the roles of both anthropogenic and natural drivers in OH trends under a warmer climate. All these models are configured with detailed interactive atmospheric chemistry and aerosols for the troposphere and stratosphere, involving the chemical system for O$_x$–HO$_x$–NO$_x$–CH$_4$–CO reactions and NMVOC oxidation.

In this study, we focus on the period of 2015–2100 using the transient CMIP6 simulations in future scenarios. Designed for the achievement of the carbon neutrality in the world by the second half of the 21st century to limit the global temperature rise within 2.0 degree, preferably below 1.5 degree, one commonly-adopted climate scenario, SSP1-26 scenario, is chosen for analysis among various possible socioeconomic futures[16,36]. The SSP1-26 represents the green road with a global shift toward a more sustainable future, aimed to achieve the radiative forcing target at 2.6 W m$^{-2}$ over the course of century to meet the UN Paris Agreement. Following this scenario, Global CO$_2$ emission will reach zero threshold around 2070, and emissions of aerosols and trace gases like sulfur dioxide and nitrogen oxides from energy and industrial sectors decline sharply due to extensive use of non-fossil fuel energy and end-of-pipe measures.

As a comparison, we examine the OH evolution and its effects on CH$_4$ in an unmitigated climate scenario with no effective climate policies carried out, called SSP3-70, which represents a high radiative forcing of 7.0 W m$^{-2}$ with carbon emission continuing to rise throughout the 21st century. Compare to other SSP scenarios, SSP3-70

shows the highest emissions of methane and other near-term climate forcers at 2100, equivalent to or exceeding the present-day levels. Of them, anthropogenic methane emission exhibits a steadily increasing trend with the 2100 level almost doubling that of 2015[20]. Overall, the SSP1-26 and SSP3-70, as two established scenarios with divergent emission trajectories, can provide a robust experimental platform to study atmospheric chemistry-climate interactions and to inform optimum climate policy formulation.

From CMIP6 models, the $CH_4$ simulations are driven by prescribed surface concentrations as lower-boundary conditions from present day to the future constructed following different scenarios. These future $CH_4$ concentrations have been projected in support of CMIP6 using a much simplified climate-carbon-cycle model without the consideration of evolution of natural $CH_4$ emissions in response to climate warming[42]. A more physically-based method is to model $CH_4$ concentrations online with emission fluxes, i.e., the emission-driven mode, which allows a fully interactive, process-level simulation of the $CH_4$ budget and its response to climate change[47], but this is beyond the scope of this study. The current CMIP6 simulations are able to reflect the changes in global OH in relation to evolving $CH_4$ concentrations given in different scenarios, but their results are not fully self-consistent, as each model simulation infers its own $CH_4$ trajectory that typically differs from the prescribed $CH_4$ trajectory for the scenario.

Of CMIP6 ensemble simulations, we chose ensemble member "r1i1p1f1" for GFDL-ESM and MRI-ESM and "r1i1p1f2" for UKESM. All models provide monthly results of atmospheric mixing ratios of various trace species including OH, $CH_4$, $O_3$ and $NO_2$, which are averaged in our analyses to obtain annual means. The diagnosed $CH_4$ loss rates in each grid box that are controlled predominantly by the OH-$CH_4$ reaction are extracted. Then, we can derive the global-mean $CH_4$ lifetime by combining its whole-atmosphere integrated loss fluxes and the global total $CH_4$ burden. We also calculate the normalized $CH_4$ loss rate ($yr^{-1}$) by dividing the $CH_4$ loss by the $CH_4$ burden.

In the CMIP6 archive, the loss fluxes of $CH_4$ and CO with OH are provided, which are used to show the changes in OH sinks, while the OH production fluxes are not available from the CMIP6 outputs. To enable the comparison of the changes between the OH sources and sinks, we combine the modeled trace gas mixing ratios and diagnosed $O_3$ photolysis rates to estimate the production and loss fluxes of OH in the troposphere. Based on the availability of CMIP6 data, we consider three source terms ($O(^1D) + H_2O$, $NO + HO_2$, and $O_3 + HO_2$) and six sink terms (the reaction of OH with $CH_4$, CO, $O_3$, $HO_2$, HCHO, and $NO_2$). It has been suggested that based on steady-state approximations, applying these terms can derive reasonable spatial and temporal information on OH at a global scale[48]. The change in water vapour concentrations is include in the calculation. The temperature dependent reaction rate coefficients are derived using the time- and grid-resolved temperature from the models. Our calculation includes a representation of the effect of global warming on the OH budget.

### Theoretical box model
The chosen CMIP6 models above can represent the effects of $CH_4$ on atmospheric chemistry including OH evolution, but they are not able to simulate $CH_4$ mixing ratios in the atmosphere interactively because $CH_4$ concentrations in the models are constrained by the surface boundary input data. In this study, to reveal the importance of OH in driving $CH_4$ evolution, we reconstruct the inter-annual variations in global mean $CH_4$ concentrations in different scenarios using a simple theoretical box[49] model expressed as the equation below:

$$\frac{d[CH_4]}{dt} = \frac{-[CH_4]}{\tau} + E \qquad (2)$$

Where $[CH_4]$ denotes the mixing ratio (ppbv) of $CH_4$ in the atmosphere; t denotes time in years; $\tau$ denotes $CH_4$ lifetime in years

calculated using the CMIP6 results and a constant lifetime (150 years) with respect to soil uptake; and E denotes the growth of $CH_4$ concentration in ppb per year associated with the emission flux. The CMIP6 database provides the model outputs of $CH_4$ loss fluxes, which are integrated in time and space in this study to obtain the annual total loss for each model. In line with Prather et al.[50], we use the $CH_4$ turnover lifetime, defined as the $CH_4$ burden divided by the loss rate, to project future global-mean $CH_4$ abundance. The lifetimes derived from transient chemistry-climate simulations consider various chemistry-climate factors in determining OH, including the $CH_4$ feedback.

In the Eq. 2, the global anthropogenic and biomass burning emissions of $CH_4$ are provided by the input data sets for Model Intercomparison Projects (input4MIPs) in accordance with CMIP6 SSP scenarios. Natural emission including wetland is taken from the World Data Center for Climate (WDCC) at DKRZ[51]. In this database, the natural emission at the year 2015 is 222 Tg. These long-term emissions are used to model $[CH_4]$ in Eq. 2 based on a constant factor of 2.75 Tg per ppbv, which is applied for the conversion from emission fluxes ($Tg\ yr^{-1}$) to atmospheric mixing ratios (ppbv)[50]. We solve this differential equation to obtain $[CH_4]$ using an explicit method. Because $CH_4$ lifetimes evolve with OH variations from 2015 to 2100, the $[CH_4]$ variability is dependent on both emissions and lifetime changes. We also obtain the results by fixing the lifetime at the 2015 level in Eq. 2 and compare them with those using varying OH to identify the role of OH in shaping global $CH_4$ concentrations in the 21st century for SSP1-26 and SSP3-70 scenarios, respectively.

### Climate forcing estimation
Using these reconstructed $CH_4$ concentrations in different cases, we then estimate the $CH_4$ radiative forcing ($W\ m^{-2}$) relative to pre-industrial levels using the following equation[52]:

$$RF(CH_4) = \left( -1.3 \times 10^{-6} \times \frac{(M + M_0)}{2} - 8.2 \times 10^{-6} \times \frac{(N + N_0)}{2} + 0.043 \right)$$
$$\times (\sqrt{M} - \sqrt{M_0})$$

$$(3)$$

Where M denote the global mean mixing ratios of $CH_4$ derived from Eq. 2; N denotes the $N_2O$ mixing ratio simulated by the CMIP6 models; $M_0$ and $N_0$ denote the initial $CH_4$ and $N_2O$ at the preindustrial (for the year 1750) level and equal to 722 ppb and 270 ppb, respectively[52]. Note that this RF is analogous to instantaneous forcing, but the former including adjustment to stratospheric temperatures. We calculate the $CH_4$ climate forcing separately with and without the perturbation due to OH by 2100. The OH projections are provided for a carbon neutrality scenario (SSP126) and a heavy emission scenario (SSP370), respectively.

### Trend analysis
We perform the time series analysis for interannual mean global OH as well as other trace species by using the Mann-Kendall non-parametric test for monotonic trend. The linear trend is calculated using the Theil-Sen robust estimate.

## Data availability
The model data from the Coupled Model Intercomparison Project Phase 6 are publicly available on the Earth System Grid Federation (ESGF) website (https://esgf-data.dkrz.de/search/cmip6-dkrz/ or https://aims2.llnl.gov/search/cmip6/). The source data for reproducing the figures are available in a public repository at: https://doi.org/10.5281/zenodo.10784591.

## Code availability
The codes created for analyzing the CMIP6 experiments are publicly available (https://doi.org/10.5281/zenodo.10820551).

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

## Acknowledgements

This study was supported by the National Natural Science Foundation of China (NSFC) (92044302, 42075180, and 91844301). H. Matsui was supported by the Ministry of Education, Culture, Sports, Science and Technology of Japan and the Japan Society for the Promotion of Science (MEXT/JSPS) KAKENHI Grant Numbers JP19H04253, JP19H05699, JP19KK0265, JP20H00196, JP20H00638, JP22H03722, JP22F22092, JP23H00515, JP23H00523, and JP23K18519 and MEXT Arctic Challenge for Sustainability phase II (ArCS-II; JPMXD1420318865) projects, and by the Environment Research and Technology Development Fund 2–2003 (JPMEERF20202003) and 2–2301 (JPMEERF20232001) of the Environmental Restoration and Conservation Agency. We would like to thank Dr. M. Prather (UCI) for helpful discussions. We acknowledge the World Climate Research Programme, which, through its Working Group on Coupled Modelling, coordinated and promoted CMIP6. We thank the climate modeling groups for producing and making available their model output (listed in Supplementary Table 1), the Earth System Grid Federation (ESGF) for archiving the data and providing access, and the multiple funding agencies who support CMIP6 and ESGF.

## Author contributions

T.Z. conceived and led the study. M.L., Y.S. and H.M. performed research and wrote the draft. M.L. Y.S., H.M., F.S., L.K., X.C., H.Z. and T.Z. interpreted the results and commented on the manuscript.

## Competing interests

The authors declare no competing interests.
