## [Peer Review File · Nature Communications]

Enhanced atmospheric oxidation toward carbon neutrality reduces methane's climate forcingREVIEWER COMMENTS

Reviewer #1 (Remarks to the Author):

Review of “Enhanced atmospheric oxidation toward carbon neutrality reduces methane’s climate forcing” by Mingxu Liu et al.

They produced an interesting analysis of the model outputs from three ocean-atmosphere coupled earth system models and contrasted the SSP126 and the SSP370 scenarios to evaluate future OH trends in response to interactive chemistry and emission changes.

The existing bibliography is already quite comprehensive. I would just recommend reading Akimoto, H, and Tanimoto, H. (2022)

The paper is well written and articulated. The methodology is sound and detailed. However, the outcomes from the simulation are not that surprising considering the scenarios and the models employed. In other words, the impact on OH is somehow expected from the emissions scenarios and the combined effects does not bring an additional synergetic effect that is counterintuitive.

References:

Akimoto, H, Tanimoto, H. 2022. Rethinking of the adverse effects of NO_x-control on the reduction of methane and tropospheric ozone—Challenges toward a denitrified society. Atmospheric Environment 277: 119033. DOI: <http://dx.doi.org/10.1016/j.atmosenv.2022.119033>.

Reviewer #2 (Remarks to the Author):

Review of “Enhanced atmospheric oxidation toward carbon neutrality reduces methane’s climate forcing”, by Mingxu Liu et al.

General Comments

This study analyses oxidant changes in future CMIP6 simulations performed by three Earth System Models, and the implications for methane changes under two future scenarios. Changes in the hydroxyl radical (OH) are found to exert a significant control on future methane levels, which is an interesting result that has not previously been analysed for these model integrations, so it is a finding well worth publishing.

However, the text is not always as clear as it could be, it misses some earlier very relevant work, and the analysis could have a bit more depth (see my suggestions below). In particular, the contribution of

changes in physical climate (temperature, water vapour, etc.) to the changes in OH (and, more specifically, the flux through CH₄+OH) should be discussed, as should the origins of the future mixing ratios of CH₄ used as the driving boundary condition in all the CMIP6 simulations analysed.

The authors seem to think that the finding that future changes in OH, driven (mainly?) by large reductions in future anthropogenic emissions of NO_x and CO (and CH₄), that exert opposing influences on OH, is “unexpected” (line 18). Maybe I am atypical, but I don’t find this too surprising – and several papers, including many referenced by the authors in their manuscript, have documented this sort of response before. So in this sense, the text as written overly extols a sense of novelty that is unjustified. The authors should tone this down, and just focus on the real novelty – i.e. that no-one has analysed the OH trends in these model’s future CMIP6 simulations, and that this reveals that a future trajectory towards carbon neutrality should have the added benefit of enhancing OH and shortening the methane lifetime.

The analysis would be strengthened if the authors could quantify and attribute the individual component drivers of the future OH (and hence CH₄) trends (i.e., CO, NO_x, CH₄ emissions, increases in surface temperature, increases in humidity, changes in surface albedo, etc.). This may not be fully possible with the limited diagnostics available in the CMIP6 archive, but I think a deeper analysis and discussion of at least some of these should be possible. If not, it would be worth documenting, so that future community modelling efforts (e.g., CMIP7) could be strengthened with the required diagnostics.

If the manuscript is adjusted in this way, I do feel it would make a good contribution to the literature.

Specific Comments

L17: “to be” – this makes it sound like reality, but it is, of course, a projection. Adjust.

L21: If you quote a percentage change in something, you have to state the reference point.

L21: “realistic” – it is impossible to say a scenario is realistic until it has actually happened (and then it is not a scenario...). Adjust.

L33: “evolves” – has evolved (in the past) and will evolve (in the future)

L40: “will be” – will be is too certain – say may or likely to.

L41: 21st Century

L48: as -> at

L57: The other significant influence on OH in these scenarios beyond their emissions that should be mentioned is the change in physical climate (temperature, water vapour, etc.). This has been explored in models for over 20 years now – e.g., early modelling efforts by Johnson et al. (2001) and others should be noted and referenced.

L61: earth should be capitalised: Earth

L63: 'ambitious' – maybe this is OK, but I would avoid including 'emotional'/subjective adjectives to describe scenarios. They are just different scenarios – I'd say high or low, but probably no more than that. I'm not a scenario person, but scenario people have told me this before.

L77-78: "inter-annual rates of 0.071-0.16% yr⁻¹" – firstly, I think this is a trend (I'm not sure what an inter-annual rate is). Secondly, it is unclear how these values relate to Figure 1d. Are they the range of model trends over 2020-2100? Or different time periods? Clarify.

L81/82: Same comment as previous for these trends.

L88-89: "likely translates into a similar decrease" – This depends on the H₂O and photolysis rates. It would be nice to see the change in the O₁D+H₂O flux. Is it in the CMIP6 output for these models?

L97: "natural driver" – of what?

L98: Reference 38 – it would be useful to reference Johnson et al (2001) (and probably others) here.

L99: Clarify this is for the SSP1-26 scenario

L101: are able to promote -> promotes

L102: "probably" – can you not be more definitive and quantitative here?

L104: The increase in water vapour will be much bigger in the SSP3-70 scenario – and thus play a bigger role in the OH change.

L119: Delete "on"

L123-127: This is an interesting idea, but needs backing up with more specific results, i.e. changes in ozone photolysis fluxes from the models. Are these in the CMIP6 output for these models?

L134: To get a total (whole atmosphere) CH₄ lifetime (which is, I think, what you need for your modelling), you should also include the CH₄ losses to Cl and stratospheric losses. See Prather et al. (2012).

L136: Johnson et al. (2001) show that the impact of changes in physical climate alone (SRES A2 in 2100, relative to 1990; surface global warming of ~3K) on CH₄ lifetime is to reduce it from 9.7 yr to 8.5 yr, i.e. shorten the lifetime by about 1.2 years. This is quite comparable to the changes you are seeing in your results, and my guess is that a significant fraction of the lifetime change that you see is coming from warming. This contribution should be more prominently discussed, and if possible, quantified (or at least estimated based on earlier studies).

L145 or thereabouts: Refer to the specified CH₄ concentrations in the official SSP scenarios (i.e. your Extended data Figure 1). These concentrations were created by an integrated assessment model that (I think) did at least try and account for changes in CH₄ lifetime over the 21st century. Exactly what they did should be clearly documented here, so that the difference with what you have done is much more explicit.

L182: “secondary importance” – How do you know the physical climate effect is secondary, since you haven’t isolated it? My guess is that the physical climate component of the OH change is a large component (although maybe not the biggest component) in your results.

L189: severing -> serving

L235: SSP3-70 represents a radiative forcing of 7.0 W m⁻²

L246: “without consideration of evolution of natural CH₄ emissions” – but do they attempt to include the evolution of NO_x and CO emissions and their impacts on CH₄ lifetime via OH?

L253: I’m slightly worried that you have used “...f1” for GFDL and MRI, but “...f2” for UKESM. I am not entirely sure what these codes mean, but thought that the “f” referred to the forcing dataset used for the simulation. If that is true, can you clarify how different f1 and f2 are? Hopefully they have the same (or very similar) CH₄ concentrations and NO_x, CO, etc. emissions.

L279: The conversion factor from Tg to ppbv has an IPCC-recommended method described in Prather et al. (2012), which I think gives a (slightly) different value. (Apologies if this is exactly what you are using).

L286: You really should compare your box model results with the CH₄ mixing ratios specified in the SSP scenarios (i.e. in the IPCC-AR6 Appendix)

L292: denotes

L293: Did the CMIP6 models simulate N₂O mixing ratios? Or did they prescribe N₂O from the scenario tables?

L460: Figure 1 – There will be some level of inconsistency for the CH₄ in these simulations that should be acknowledged and quantified. The SSP126 scenario specifies CH₄ concentrations (as shown in Extended data Figure 1). Since each model simulates different values of OH, they will also simulate different lifetimes. Hence the inferred CH₄ emissions will differ for each model in the future. They could be calculated and compared. At the very least, this inherent inconsistency, brought about by models using prescribed CH₄ mixing ratios determined by a different model, should be carefully noted and discussed, since it is central to the research presented.

The units of Figure 1e should be Tmol/yr (or per some unit of time)?

L475: Figure 2 – Do you mean increases in the ratio of NO_x to CO mixing ratios? (i.e. delta (NO_x/CO)); or

do you mean the ratio of the increases? (i.e. $(\Delta \text{NO}_x)/(\Delta \text{CO})$).

L482: Figure 3 – I suggest include the official SSP scenario CH₄ changes on panels (b) and (c). They provide a useful reference point for the model changes shown.

L490: Figure 4 – Again, I suggest include a comparison to the official IPCC radiative forcing values for these scenarios as a useful reference. It should be noted that all the radiative forcing values are relative to pre-industrial (1750?).

L508: rations -> ratios

L521: Extended Data Figure 5: The colour scale should be improved, so that the 0 contour is more obvious (suggest use red for increases, blue for decreases, fading to white at 0% - i.e. like Extended Data Figure 6).

References

Johnson, C. E., et al. (2001), Role of climate feedback on methane and ozone studied with a Coupled Ocean-Atmosphere-Chemistry Model, *Geophys. Res. Lett.*, vol. 28, 9, 1723-1726, doi:10.1029/2000GL011996.

Prather, M. J., Holmes, C. D., and Hsu, J. (2012), Reactive greenhouse gas scenarios: Systematic exploration of uncertainties and the role of atmospheric chemistry, *Geophys. Res. Lett.*, 39, L09803, doi:10.1029/2012GL051440.

Reviewer #3 (Remarks to the Author):

In "Enhanced atmospheric oxidation toward carbon neutrality reduces methane's climate forcing" the authors present results from an analysis CMIP6 of model data with a focus on the atmospheric methane trend under two of the Shared Socio-Economic Pathways, namely SSP1-2.6 and SSP3-7.0. SSP1-2.6 is used as a proxy for the pathway to carbon neutrality. The novel idea in this study is that mitigation of the co-emitted NTCFs, such as NO_x, CO, VOC and aerosols will also impact on the OH abundance in the troposphere, thereby modulating the CH₄+OH lifetime. CO and VOC directly compete with methane for OH and NO_x, CO, and VOCs are important ozone precursors and are expected to impact the ozone abundance, ultimately affecting OH in the course. The mitigation of co-emitted NTCFs under SSP1-2.6 is shown to lead to a further substantial reduction in methane with consequences for RF and GMST.

The principal idea of this study is novel and original, as far as I am aware. The effect, based on the analysis of output from three CMIP6 Earth system models, could be quite substantial, both in unexpected additional reductions in the methane RF under a carbon-neutral pathway and in further

negative surprises via an increased methane RF under a heavy emission pathway. In my opinion, this is an important contribution to a current hot topic, the mitigation of methane RF. For this reason and for the potential impact on policy making I support the publication of this study.

The choice of methods is robust and the presentation of results is also done fairly well, but could improved, perhaps. I don't see a principal problem with scientific methods, data analysis and presentation.

The data and code availability section needs to be improved, I think. The paper needs a separate, dedicated DOI repository with the data and code used to produce this paper, not just the output from the CMIP6 models. Whatever was used, python scripts, Excel spreadsheets, any other code or data directly linked to this study needs to be made available for inspection. At least that is the way how I understand Data and Code availability.

Where the current manuscript needs major improvement the text itself. I appreciate that the author's first language may not be English (neither is it mine). But the authors need to make a serious effort in improving the text.

Therefore, I recommend publication after major revisions.

Detailed comments can be found in the attached PDF-document. Some of my suggestions may help in directing the necessary revisions.

Response to reviewers' comments

Reviewer #1 (Remarks to the Author):

Review of “Enhanced atmospheric oxidation toward carbon neutrality reduces methane’s climate forcing” by Mingxu Liu et al.

Response: We would like to appreciate the reviewer’s valuable comments. Please see our detailed responses and revisions below. The comments are marked in blue color, followed by our responses.

They produced an interesting analysis of the model outputs from three ocean-atmosphere coupled earth system models and contrasted the SSP126 and the SSP370 scenarios to evaluate future OH trends in response to interactive chemistry and emission changes. The existing bibliography is already quite comprehensive. I would just recommend reading Akimoto, H, and Tanimoto, H. (2022)

Response: Accepted. This reference is added in our revised manuscript. Please see the citation at Line 46-47. In this review paper, Akimoto and Tanimoto (2022) summarize the comprehensive effects of NO_x controls on atmospheric oxidation, methane, and ozone levels. Their paper can support the significance of our study to quantify the future OH trends linked to projected emission reductions of NO_x and reactive carbon species.

The paper is well written and articulated. The methodology is sound and detailed. However, the outcomes from the simulation are not that surprising considering the scenarios and the models employed. In other words, the impact on OH is somehow expected from the emissions scenarios and the combined effects does not bring an additional synergetic effect that is counterintuitive.

Response: Considering the reviewer’s concern, we would like to point out the following.

- 1) This study demonstrates that in the future climate-mitigation scenario (i.e., carbon neutrality) of CMIP6 simulations, the reduction in the strength of OH sinks outweighs that of OH sources, consequently yielding the enhancement of atmospheric oxidation. To our knowledge, this finding has not been reported to date. Please see our detailed analysis at Line 89-120 in the revised manuscript.
- 2) The OH trend projections in previous studies (e.g., Holmes et al., 2013 and Stevenson et al., 2013) were based on RCP scenarios and earlier versions of CMIP. We show the first analysis of future OH drivers and effects in the CMIP6 dataset (the reference data for IPCC AR6 report). Please see Line 51-58 in the revised manuscript.

3) This study provides a quantitative understanding of OH trends and drivers in the future world based on ensemble climate model simulations. The results are important for the accurate projection of near-term climate forcings and can guide the formulation of optimal climate policies. More in-depth analysis results have also been added in our revised manuscript based on other two reviewers' comments.

References:

Akimoto, H, Tanimoto, H. 2022. Rethinking of the adverse effects of NO_x-control on the reduction of methane and tropospheric ozone—Challenges toward a denitrified society. *Atmospheric Environment* 277: 119033. DOI: <http://dx.doi.org/10.1016/j.atmosenv.2022.119033>.

Holmes, D., M. J. Prather, O. A. Søvde, G. Myhre, Future methane, hydroxyl, and their uncertainties: key climate and emission parameters for future predictions. *Atmos. Chem. Phys.* 13, 285-302 (2013).

Stevenson, S., P. J. Young, V. Naik, J. F. Lamarque, D. T. Shindell, A. Voulgarakis, R. B. Skeie, S. B. Dalsoren, G. Myhre, T. K. Berntsen, G. A. Folberth, S. T. Rumbold, W. J. Collins, I. A. MacKenzie, R. M. Doherty, G. Zeng, T. P. C. van Noije, A. Strunk, D. Bergmann, P. Cameron-Smith, D. A. Plummer, S. A. Strode, L. Horowitz, Y. H. Lee, S. Szopa, K. Sudo, T. Nagashima, B. Josse, I. Cionni, M. Righi, V. Eyring, A. Conley, K. W. Bowman, O. Wild, A. Archibald, Tropospheric ozone changes, radiative forcing and attribution to emissions in the Atmospheric Chemistry and Climate Model Intercomparison Project (ACCMIP). *Atmos. Chem. Phys.* 13, 3063-3085 (2013).

Reviewer #2 (Remarks to the Author):

Review of “Enhanced atmospheric oxidation toward carbon neutrality reduces methane’s climate forcing”, by Mingxu Liu et al.

General Comments

This study analyses oxidant changes in future CMIP6 simulations performed by three Earth System Models, and the implications for methane changes under two future scenarios. Changes in the hydroxyl radical (OH) are found to exert a significant control on future methane levels, which is an interesting result that has not previously been analyzed for these model integrations, so it is a finding well worth publishing.

Response: We appreciated the reviewer’s helpful comments and accepted ALL of them. The comments are marked in blue color, followed by our revisions.

However, the text is not always as clear as it could be, it misses some earlier very relevant work, and the analysis could have a bit more depth (see my suggestions below). In particular, the contribution of changes in physical climate (temperature, water vapour, etc.) to the changes in OH (and, more specifically, the flux through CH₄+OH) should be discussed, as should the origins of the future mixing ratios of CH₄ used as the driving boundary condition in all the CMIP6 simulations analysed.

Response: Accepted. Following the suggestions below, we revised our manuscript by adding more in-depth analysis, including:

- 1) The detailed budget analysis of global OH is added. We quantify the changes in OH sources and sinks due to changes in precursor emissions and physical climate (mainly temperature and water vapor). Please see our added analysis at Line 89-120 and Line 152-161 in the revised manuscript.
- 2) The discussion on the use of prescribed CH₄ data as the boundary condition of CMIP6 simulations is added. The prescribed CH₄ data for CMIP6 was derived from a simple model projection using a linear equation for OH variation (MAGICC7.0; see Meinshausen et al., 2020). That predicted CH₄ decrease by 2100 for the SSP126 scenario is generally lower than our estimates of CH₄ concentrations using the same CH₄ emissions but with diagnostic lifetimes from the complex climate models (Supplementary Fig. 7). As detailed atmospheric chemical processes are lacked in the reduced-complexity climate model, the prescribed CH₄ trends used for CMIP6 projections need to be revisited. Please see Line 178-192.

Please see our detailed revisions in the following responses.

The authors seem to think that the finding that future changes in OH, driven (mainly?) by large reductions in future anthropogenic emissions of NO_x and CO (and CH₄), that exert opposing influences on OH, is “unexpected” (line 18). Maybe I am atypical, but I don’t find this too surprising – and several papers, including many referenced by the authors in their manuscript, have documented this sort of response before. So in this sense, the text as written overly extols a sense of novelty that is unjustified. The authors should tone this down, and just focus on the real novelty – i.e. that no-one has analysed the OH trends in these model’s future CMIP6 simulations, and that this reveals that a future trajectory towards carbon neutrality should have the added benefit of enhancing OH and shortening the methane lifetime.

Response: Accepted. Please see the revisions below.

- 1) At Line 18, we delete the word “unexpected” and rewrite the sentence as “The leading cause of this OH enhancement is”.
- 2) We also add the sentence suggested by the reviewer, that is, “We find that a future trajectory towards carbon neutrality has the added benefit of CH₄ mitigation with.” Please see Line 219-220 in our revised manuscript.

The analysis would be strengthened if the authors could quantify and attribute the individual component drivers of the future OH (and hence CH₄) trends (i.e., CO, NO_x, CH₄ emissions, increases in surface temperature, increases in humidity, changes in surface albedo, etc.). This may not be fully possible with the limited diagnostics available in the CMIP6 archive, but I think a deeper analysis and discussion of at least some of these should be possible. If not, it would be worth documenting, so that future community modelling efforts (e.g., CMIP7) could be strengthened with the required diagnostics.

Response: Accepted. A deeper analysis of the component drivers is added in our revised manuscript, as summarized below.

- 1) Based on the available CMIP6 archive, the strengths of chemical sources (O(¹D)+H₂O, NO+HO₂, etc.) and sinks (OH+CO, OH+CH₄, OH+O₃, OH+HCHO, etc.) of OH are estimated and then compared to find their net effect on OH levels. We show that the decreases in OH sources are smaller than the decreases in OH sinks, resulting in OH enhancement across models. Please see Line 89-120.
- 2) The effects of physical climate changes including temperature and humidity on OH and CH₄ trends are quantified and discussed. Temperature increases (i.e., global warming) can increase the reaction rate of CH₄ with OH and therefore shorten the CH₄ lifetime. In our examined case, i.e., SSP1-26 (low forcing scenario), the global-mean surface temperature increases from 2015 to 2100 are only 0.53K–1.6K across models. We estimate that the CH₄

lifetimes are shortened by 0.17–0.30 years across models due to increased temperature alone. The corresponding values due to OH increase are 0.61–1.7 years. The temperature effect is therefore important, though not the dominant driver. The water vapour increase enhances the OH production by 3.4–4.3 Tmol yr⁻¹, which is much lower than the reduction of OH sinks (59–64 Tmol yr⁻¹), such the water vapor effect is minor. Please see these revisions at Line 95-103 and Line 152-161.

If the manuscript is adjusted in this way, I do feel it would make a good contribution to the literature.

Response: Accepted. All the comments raised by the reviewer are addressed and the manuscript is revised accordingly.

Specific Comments

L17: “to be” – this makes it sound like reality, but it is, of course, a projection. Adjust.

Response: Accepted. We reword it to “projected to”. Please see Line 16 in the revised manuscript.

L21: If you quote a percentage change in something, you have to state the reference point.

Response: Accepted. We remove the percentage values here. Please see Line 20.

L21: “realistic” – it is impossible to say a scenario is realistic until it has actually happened (and then it is not a scenario...). Adjust.

Response: Accepted. We change it to “largely unmitigated scenario”, which is often used to describe SSP3-70 scenario. Please see Line 20-21.

L33: “evolves” – has evolved (in the past) and will evolve (in the future)

Response: Accepted. We reword it to “will evolve”, as this study focuses on OH prediction in the 21st Century. Please see Line 32.

L40: “will be” – will be is too certain – say may or likely to.

Response: Accepted. We reword it to “is likely”. Please see Line 39.

L41: 21st Century

Response: Accepted. We reword it to “21st Century”. Please see Line 40.

L48: as -> at

Response: Accepted. We reword it to “at”. Please see Line 48.

L57: The other significant influence on OH in these scenarios beyond their emissions that should be mentioned is the change in physical climate (temperature, water vapour, etc.). This has been explored in models for over 20 years now – e.g., early modelling efforts by Johnson et al. (2001) and others should be noted and referenced.

Response: Accepted. We add references Johnson et al. (2001) and others (Johnson et al., 1999) here and describe the roles of physical climate in shaping OH trends. Please see Line 48-50 and Line 57-58.

L61: earth should be capitalised: Earth

Response: Accepted. We rewrite it to “Earth”. Please see Line 62.

L63: ‘ambitious’ – maybe this is OK, but I would avoid including ‘emotional’/subjective adjectives to describe scenarios. They are just different scenarios – I’d say high or low, but probably no more than that. I’m not a scenario person, but scenario people have told me this before.

Response: Accepted. The word ‘ambitious’ is removed in the description of the scenario throughout the text. Instead, we use the word ‘strong climate mitigation’ or ‘low-forcing’ for that. Please see Line 64.

L77-78: “inter-annual rates of 0.071-0.16% yr⁻¹” – firstly, I think this is a trend (I’m not sure what an inter-annual rate is). Secondly, it is unclear how these values relate to Figure 1d. Are they the range of model trends over 2020-2100? Or different time periods? Clarify.

Response: Accepted. This is a trend analysis. The values are the ranges of model results over 2015–2100. We clarify them in the revised manuscript. Please see Line 78-79.

L81/82: Same comment as previous for these trends.

Response: Accepted. We clarify this is a trend over 2015-2100. Please see Line 81-82.

L88-89: “likely translates into a similar decrease” – This depends on the H₂O and photolysis rates. It would be nice to see the change in the O(1D)+H₂O flux. Is it in the CMIP6 output for these models?

Response: Accepted. We provide quantitative results here. The changes in the O(1D)+H₂O flux is not available from the CMIP6 output, but the production rate of O(1D) is presented. We therefore use photolysis rate of O₃ to produce O(1D) together with H₂O concentration to estimate the OH primary production (O(1D)+H₂O) flux and its changes. The results show that due to reduction of global O₃ burden in the SSP126 scenario, at 11-27% across models, the global OH primary production is reduced by 16-22%. Such analysis is added in our revised manuscript at Line 89-95. The method to this estimation is added at Line 286-297.

L97: “natural driver” – of what?

Response: Accepted. This should be “natural driver of OH abundance”. The sentence is rewritten. Please see Line 95-96.

L98: Reference 38 – it would be useful to reference Johnson et al (2001) (and probably others) here.

Response: Accepted. The references Johnson et al. (2001) is added here. Please see Line 95-96.

L99: Clarify this is for the SSP1-26 scenario

Response: Accepted. We add the statement that this is for the SSP1-16 scenario. Please see Line 98.

L101: are able to promote -> promotes

Response: Accepted. This description is removed in our revised manuscript.

L102: “probably” – can you not be more definitive and quantitative here?

Response: Accepted. The quantitative result is added here. By conducting the budget

analysis of global OH, we make it definitive that the suppression of OH sink consistently outweighs that of OH sources in each model projection. Please see Line 116-118 and Supplementary Fig. 5.

L104: The increase in water vapour will be much bigger in the SSP3-70 scenario – and thus play a bigger role in the OH change.

Response: Accepted. Here, we state the result is for the SSP1-26 scenario. In the SSP3-70 scenario, the rise in CH₄ concentration leads to OH decrease, though the increase in water vapour partly offsets this impact. Please see Line 95-102.

L119: Delete “on”

Response: Accepted. We reword it to “With regards to”. Please see Line 134.

L123-127: This is an interesting idea, but needs backing up with more specific results, i.e. changes in ozone photolysis fluxes from the models. Are these in the CMIP6 output for these models?

Response: Accepted. The reduction of surface albedo might contribute to OH decrease over the Arctic surface. However, we also agree with Reviewer #3 that the Arctic photochemistry is extremely complex than presented here. Our study focuses on global-mean OH changes, not specifically on Arctic OH. As suggested by Reviewer #3, we remove this analysis in the revised manuscript.

L134: To get a total (whole atmosphere) CH₄ lifetime (which is, I think, what you need for your modelling), you should also include the CH₄ losses to Cl and stratospheric losses. See Prather et al. (2012).

Response: Accepted. In the CMIP6 model output, the variable “lossch4” is provided, which means the total atmospheric sink due to its reaction with OH and Cl as well as the stratospheric losses. In other words, these loss pathways have been included in our analysis. Our original description was inaccurate, so we rewrite this sentence to clarify that we consider the total CH₄ lifetime. Please see Line 139-143 in our revised manuscript.

L136: Johnson et al. (2001) show that the impact of changes in physical climate alone (SRES A2 in 2100, relative to 1990; surface global warming of ~3K) on CH₄ lifetime is to reduce it from 9.7 yr to 8.5 yr, i.e. shorten the lifetime by about 1.2 years. This is quite

comparable to the changes you are seeing in your results, and my guess is that a significant fraction of the lifetime change that you see is coming from warming. This contribution should be more prominently discussed, and if possible, quantified (or at least estimated based on earlier studies).

Response: Accepted. Following the reviewer's suggestion, the impacts of changes in physical climate on CH₄ lifetime are quantified below.

1) Temperature increases (i.e., global warming) can increase the reaction rate of CH₄ with OH and therefore shorten the CH₄ lifetime. In our examined case, i.e., SSP1-26 (low forcing scenario), the global-mean surface temperature increases from 2015 to 2100 are only 0.53K–1.6K across models. Based on the grid-resolved temperature changes from 2015 to 2100 and the CH₄ and OH concentrations from the CMIP6 outputs, we estimate that the CH₄ lifetimes are shortened by 0.17–0.30 years across models due to increased temperature alone. The corresponding values due to OH increase are 0.61–1.7 years. The temperature effect is therefore important, though not the dominant driver. We add such analysis related to the negative climate feedback on the CH₄ cycle at Line 152-161 in the revised manuscript.

2) Water vapour increase can reduce CH₄ lifetime through increase in OH primary production. Such effect has been quantified in our manuscript. We find that compared to emissions reductions, the effect of water vapour increase is minor on global OH change. Please see Line 95-103 in the revised manuscript.

L145 or thereabouts: Refer to the specified CH₄ concentrations in the official SSP scenarios (i.e. your Extended data Figure 1). These concentrations were created by an integrated assessment model that (I think) did at least try and account for changes in CH₄ lifetime over the 21st century. Exactly what they did should be clearly documented here, so that the difference with what you have done is much more explicit.

Response: Accepted. Based on the comment, we revise our manuscript as following.

1) The SSP-specified CH₄ trends are added in Supplementary Fig. 7 in the revised manuscript. This is for comparison between the prescribed data and our model estimates using the same anthropogenic and natural CH₄ emissions.

2) The CH₄ forcing data in the SSP was created using the reduced-complexity climate model (MAGICC7.0; see Meinshausen et al., 2011). The model simply considers the CH₄ lifetime changes (mainly through OH changes) using a linear parameterization that depends on precursor emission changes. Based on the SSP1-26 emissions in 2015–2100, we estimate that this simple parameterization yields a modest increase of 7.3% in global OH, which approaches the lower end of the results (6.2%–15% over 2015–2100) from the complex chemistry-climate models in the CMIP6. This suggests that we need to revisit the prescribed CH₄ variations used as the driving boundary condition in the CMIP6 simulations. Please see the added discussion at Line 178-192.

Supplementary Fig. 7. The comparison of the CH₄ trends derived in this study with the CMIP6 prescribed data. The panels show the percentage changes relative to the 2015 levels for the (a) SSP126 scenario and (b) the SSP370 scenario. For our results, the solid and dashed lines stand for the calculation using varying OH and fixed OH, respectively. The same CH₄ emissions are used for deriving our results and the prescribed data.

L182: “secondary importance” – How do you know the physical climate effect is secondary, since you haven’t isolated it? My guess is that the physical climate component of the OH change is a large component (although maybe not the biggest component) in your results.

Response: Accepted. Following the suggestion, we rewrite this sentence as “The changes in atmospheric temperature and water vapour concentrations under a warmer climate also participate in this chemical coupling.”. Please see Line 218-219.

L189: severing -> serving

Response: Accepted. This sentence is removed.

L235: SSP3-70 represents a radiative forcing of 7.0 W m⁻²

Response: Accepted. We correct it as “radiative forcing of 7.0 W m⁻²”. Please see Line 262 in the revised manuscript.

L246: “without consideration of evolution of natural CH₄ emissions” – but do they attempt

to include the evolution of NO_x and CO emissions and their impacts on CH₄ lifetime via OH?

Response: Accepted. As mentioned in the proceeding responses, they used a simple linear equation to include the effect of OH evolution due to NO_x and CO emissions changes on CH₄ lifetime (Meinshausen et al., 2020). Their derived OH and CH₄ trends are different from the complex climate simulation results (please see Supplementary Fig. 7). We add more description here to show the source of prescribed CH₄ concentrations. Please see Line 178-192.

L253: I'm slightly worried that you have used "...f1" for GFDL and MRI, but "...f2" for UKESM. I am not entirely sure what these codes mean, but thought that the "f" referred to the forcing dataset used for the simulation. If that is true, can you clarify how different f1 and f2 are? Hopefully they have the same (or very similar) CH₄ concentrations and NO_x, CO, etc. emissions.

Response: Accepted. The "f" refers to forcing index. For UKESM, f1 is not used and f2 is the primary variant (Sellar et al., 2020). The same CH₄ concentration and other species emissions were used in all CMIP6 model simulations (please check "3. Forcing data sets" from the link: <https://pcmdi.llnl.gov/CMIP6/Guide/modelers.html>).

L279: The conversion factor from Tg to ppbv has an IPCC-recommended method described in Prather et al. (2012), which I think gives a (slightly) different value. (Apologies if this is exactly what you are using).

Response: Accepted. The value in Prather et al. (2012) is exactly what we are using. The conversion factor is 2.75. This value was given in Table S2 of Prather et al. (2012).

L286: You really should compare your box model results with the CH₄ mixing ratios specified in the SSP scenarios (i.e. in the IPCC-AR6 Appendix)

Response: Accepted. Please see Supplementary Fig. 7. The orange lines indicate the prescribed CH₄ used in CMIP6 for the SSP scenarios.

L292: denotes

Response: Accepted. We reword it. Please see Line 306.

L293: Did the CMIP6 models simulate N₂O mixing ratios? Or did they prescribe N₂O

from the scenario tables?

Response: Accepted. Like CH₄, they simulated N₂O mixing ratios using the prescribed N₂O concentrations as boundary conditions (Meinshausen et al., 2020). We describe this at Line 327.

L460: Figure 1 – There will be some level of inconsistency for the CH₄ in these simulations that should be acknowledged and quantified. The SSP126 scenario specifies CH₄ concentrations (as shown in Extended data Figure 1). Since each model simulates different values of OH, they will also simulate different lifetimes. Hence the inferred CH₄ emissions will differ for each model in the future. They could be calculated and compared. At the very least, this inherent inconsistency, brought about by models using prescribed CH₄ mixing ratios determined by a different model, should be carefully noted and discussed, since it is central to the research presented.

Response: Accepted. Following the reviewer's comments, we emphasize the points below in the revised manuscript. Please see our added analysis and discussion at Line 178-192.

1) The CMIP6 models used a concentration-driven mode to simulate CH₄. This can eliminate the inter-model difference of CH₄ simulations, but cannot reflect the dynamic response of CH₄ to tropospheric chemistry, though this has been roughly considered in the simple climate model (MAGICC, used to produce CH₄ prescribed data). The CMIP7 is encouraged to provide the inter-model comparison of CH₄ simulations using the emission-driven mode.

2) The inherent inconsistency for the CH₄ simulations can be inferred from modelled OH magnitude and trends. The prescribed CH₄ concentration is 1,018 ppb at the year 2100, while the estimated CH₄ lifetime range from 7.0 to 9.5 years across models. If this is the steady state concentration, the inferred CH₄ emission range from 300 Tg yr⁻¹ to 390 Tg yr⁻¹. In our modelling analysis, we also show that the declining trends in CH₄ burden are different among models; those with higher OH levels (shorter CH₄ lifetime) have larger percent decreases in CH₄ concentrations during 2015-2100 (Please see Figure 3).

The units of Figure 1e should be Tmol/yr (or per some unit of time)?

Response: Accepted. The unit should Tmol yr⁻¹. Figure 1e is revised.

L475: Figure 2 – Do you mean increases in the ratio of NO_x to CO mixing ratios? (i.e. delta (NO_x/CO)); or do you mean the ratio of the increases? (i.e. (delta NO_x)/(delta CO)).

Response: Accepted. We clarify that this is increases in the ratio of NO_x to CO mixing ratios.

L482: Figure 3 – I suggest include the official SSP scenario CH₄ changes on panels (b) and (c). They provide a useful reference point for the model changes shown.

Response: Accepted. We add a new figure, i.e., Supplementary Fig. 7, to show the comparison of CH₄ changes from the official SSP scenarios with the modeled changes, by using the same CH₄ emission between them. Adding the prescribed CH₄ changes directly to Figure 3 may mislead the readers, because the modeled changes shown in Figure 3 use different natural CH₄ emissions with the prescribed data. The projected CH₄ concentrations depend on both lifetimes and emissions.

L490: Figure 4 – Again, I suggest include a comparison to the official IPCC radiative forcing values for these scenarios as a useful reference. It should be noted that all the radiative forcing values are relative to pre-industrial (1750?).

Response: Accepted. The official IPCC AR6 radiative forcing values for the present (2019) and future (2100) are added in Figure 4 as a reference. Consistent with IPCC AR6 report, our calculations provide the radiative forcing values relative to 1750.

L508: rations -> ratios

Response: Accepted. We reword it. Please see the caption of Supplementary Fig. 2.

L521: Extended Data Figure 5: The colour scale should be improved, so that the 0 contour is more obvious (suggest use red for increases, blue for decreases, fading to white at 0% - i.e. like Extended Data Figure 6).

Response: This figure is removed, as the corresponding analysis is removed from the main text.

References

Johnson, C.E., W. J. Collins, D. S. Stevenson, R. G. Derwent, (1999) Relative roles of climate and emissions changes on future tropospheric oxidant concentrations. *J. Geophys. Res.-Atmos* 104, 18631-18645.

Johnson, C. E., et al. (2001), Role of climate feedback on methane and ozone studied with

a Coupled Ocean-Atmosphere-Chemistry Model, *Geophys. Res. Lett.*, vol. 28, 9, 1723-1726, doi:10.1029/2000GL011996.

M. Meinshausen, Z. R. J. Nicholls, J. Lewis, M. J. Gidden, E. Vogel, M. Freund, U. Beyerle, C. Gessner, A. Nauels, N. Bauer, J. G. Canadell, J. S. Daniel, A. John, P. B. Krummel, G. Luderer, N. Meinshausen, S. A. Montzka, P. J. Rayner, S. Reimann, S. J. Smith, M. van den Berg, G. J. M. Velders, M. K. Vollmer, R. H. J. Wang. (2020) The shared socio-economic pathway (SSP) greenhouse gas concentrations and their extensions to 2500. *Geosci. Model Dev.* 13, 3571-3605.

Prather, M. J., Holmes, C. D., and Hsu, J. (2012), Reactive greenhouse gas scenarios: Systematic exploration of uncertainties and the role of atmospheric chemistry, *Geophys. Res. Lett.*, 39, L09803, doi:10.1029/2012GL051440.

Sellar, A. A., Walton, J., Jones, C. G., Wood, R., Abraham, N. L., Andrejczuk, M., et al. (2020). Implementation of U.K. Earth system models for CMIP6. *Journal of Advances in Modeling Earth Systems*, 12, e2019MS001946. <https://doi.org/10.1029/2019MS001946>

Reviewer #3 (Remarks to the Author):

The reviewer's comments are marked in blue, followed by the authors' responses.

In "Enhanced atmospheric oxidation toward carbon neutrality reduces methane's climate forcing" the authors present results from an analysis CMIP6 of model data with a focus on the atmospheric methane trend under two of the Shared Socio-Economic Pathways, namely SSP1-2.6 and SSP3-7.0. SSP1-2.6 is used as a proxy for the pathway to carbon neutrality. The novel idea in this study is that mitigation of the co-emitted NTCFs, such as NO_x, CO, VOC and aerosols will also impact on the OH abundance in the troposphere, thereby modulating the CH₄+OH lifetime. CO and VOC directly compete with methane for OH and NO_x, CO, and VOCs are important ozone precursors and are expected to impact the ozone abundance, ultimately affecting OH in the course. The mitigation of co-emitted NTCFs under SSP1-2.6 is shown to lead to a further substantial reduction in methane with consequences for RF and GMST.

The principal idea of this study is novel and original, as far as I am aware. The effect, based on the analysis of output from three CMIP6 Earth system models, could be quite substantial, both in unexpected additional reductions in the methane RF under a carbon-neutral pathway and in further negative surprises via an increased methane RF under a heavy emission pathway. In my opinion, this is an important contribution to a current hot topic, the mitigation of methane RF.

For this reason and for the potential impact on policy making I support the publication of this study.

The choice of methods is robust and the presentation of results is also done fairly well, but could improved, perhaps. I don't see a principal problem with scientific methods, data analysis and presentation.

Response: Accepted. We really appreciate the reviewer's valuable comments and help with our manuscript writing. All of them are accepted and the manuscript are revised accordingly. Please see our responses below and in the attached file.

The data and code availability section needs to be improved, I think. The paper needs a separate, dedicated DOI repository with the data and code used to produce this paper, not just the output from the CMIP6 models. Whatever was used, python scripts, Excel spreadsheets, any other code or data directly linked to this study needs to be made available for inspection. At least that is the way how I understand Data and Code availability.

Response: Accepted. The codes (NCL) used to analyze the CMIP6 output and the data

produced from them are publicly available from the link:
<https://github.com/mliucn/CMIP6-SSP.git>. Please see the code availability statement.

Where the current manuscript needs major improvement the text itself. I appreciate that the author's first language may not be English (neither is it mine). But the authors need to make a serious effort in improving the text.

Response: Accepted. We improve the English language throughout the manuscript, including grammar, spelling, and expression.

Therefore, I recommend publication after major revisions.

Detailed comments can be found in the attached PDF-document. Some of my suggestions may help in directing the necessary revisions.

Response: Accepted. We followed all the comments in the attached PDF file to revise the manuscript. Please check our point-to-point responses to the comments in the attached PDF.

REVIEWER COMMENTS

Reviewer #2 (Remarks to the Author):

Review of revised version of “Enhanced atmospheric oxidation toward carbon neutrality reduces methane’s climate forcing” by Liu et al.

General comments

The authors have done a good job at responding to reviewer’s comments and I think the manuscript is improved and more or less ready for acceptance. However, please consider the following points first. I think some important points about the influence of climate change on OH, and the consequences of the CMIP6 models using prescribed CH₄ concentrations, should be more clearly stated.

I’m also a little worried that the correct lifetime to use in the box model projections should be the perturbation lifetime, not the turnover time. I’m 90% sure this is correct. The perturbation lifetime is the turnover lifetime multiplied by the CH₄ feedback factor (about 1.3), so an increase of about 30%. I think this would substantially change the main results. Please double check which lifetime is appropriate for your model and carefully justify. Apologies for not spotting this in my earlier review. Michael Prather would be the best person to ask about this.

Specific comments

L159 NB The influence of temperatures on OH via the temperature-dependent rate coefficient for CH₄+OH will (obviously) be larger in scenarios with more warming, such as SSP370 (which is also discussed in this paper), as compared to SSP126 which is presented here.

L173 Same point as previous one. The impacts of temperature and water vapour will be larger in the SSP370 scenario. These will tend to reduce the longer lifetime related to composition changes. I think this should be more clearly noted.

L186-188 “Since those models simulate a wide range of OH values and corresponding CH₄ lifetimes (Fig. 3a), the inferred CH₄ emissions will differ in the future.”

I think you can (and should) go slightly further with this interpretation. E.g., consider the UKESM results under the SSP126 scenario in Figure S7, which show a CH₄ change of over -50% in 2100. These results were derived using the CMIP6 prescribed CH₄ values (the orange line in Fig. S7), i.e. about -40% in 2100. If UKESM had used emission-driven CH₄ (with SSP126 CH₄ emissions), its CH₄ concentration in 2100 would have been even lower. So the results presented here will underestimate the true changes in methane. This is important to clearly state.

L276-277 “The current CMIP6 simulations are able to reflect the changes in global OH in relation to evolving CH₄ concentrations given in different scenarios.”

Suggest append:

but are not fully self-consistent, as each model simulation infers a different CH₄ trajectory that typically differs from the prescribed CH₄ trajectory for the scenario.

L294 Delete "Besides,"

L296 "can represent" -> includes a representation of

L307 " τ denotes CH₄ lifetime in years". I wonder if you should be using the perturbation lifetime here, not the overturning time. See Holmes 2018 (ref 42). This will actually make a rather large difference. Please check.

Reviewer #2 (Remarks on code availability):

Sorry, I haven't checked the code.

Reviewer #3 (Remarks to the Author):

In "Enhanced atmospheric oxidation toward carbon neutrality reduces methane's climate forcing" the authors present results from an analysis CMIP6 of model data with a focus on the atmospheric methane trend under two of the Shared Socio-Economic Pathways, namely SSP1-2.6 and SSP3-7.0. SSP1-2.6 is used as a proxy for the pathway to carbon neutrality. The novel idea in this study is that mitigation of the co-emitted NTCFs, such as NO_x, CO, VOC and aerosols will also impact on the OH abundance in the troposphere, thereby modulating the CH₄+OH lifetime. CO and VOC directly compete with methane for OH and NO_x, CO, and VOCs are important ozone precursors and are expected to impact the ozone abundance, ultimately affecting OH in the course. The mitigation of co-emitted NTCFs under SSP1-2.6 is shown to lead to a further substantial reduction in methane with consequences for RF and GMST.

This review refers to the revised text. The authors have addressed all of my concerns adequately and in full. The same is true for the other reviewers' concerns, at least as far as I can tell. I think this represents a major revision of the manuscript, and it has improved significantly from its previous version. In view of all this I recommend publication in the current form.

Response to reviewer's comments

Reviewer #2 (Remarks to the Author):

Review of revised version of “Enhanced atmospheric oxidation toward carbon neutrality reduces methane’s climate forcing” by Liu et al.

General comments

The authors have done a good job at responding to reviewer’s comments and I think the manuscript is improved and more or less ready for acceptance. However, please consider the following points first. I think some important points about the influence of climate change on OH, and the consequences of the CMIP6 models using prescribed CH₄ concentrations, should be more clearly stated.

Response: We appreciate the reviewer’s further comments. We have addressed all of them and accordingly improved our manuscript. Please see our detailed responses and revisions below.

I’m also a little worried that the correct lifetime to use in the box model projections should be the perturbation lifetime, not the turnover time. I’m 90% sure this is correct. The perturbation lifetime is the turnover lifetime multiplied by the CH₄ feedback factor (about 1.3), so an increase of about 30%. I think this would substantially change the main results. Please double check which lifetime is appropriate for your model and carefully justify. Apologies for not spotting this in my earlier review. Michael Prather would be the best person to ask about this.

Response: Accepted. We carefully check it and confirm that the turnover time (i.e., the budget lifetime) is appropriate for our model calculation, based on the following reasons.

1) Applying turnover time is physically correct in the calculation. The CH₄ burden is a function of the emission and loss fluxes. Our equation (Eq.1) is in line with that given in M. Prather (1994), i.e., $d[CH_4]/dt = S_{CH_4} - L_{CH_4}$. The L_{CH_4} (loss flux) stands for the CH₄ burden divided by its turnover lifetime. M. Prather et al. (2012) used the turnover lifetime to project future CH₄ abundance (Fig. 1 in the paper) with prescribed emissions, the methods followed by our study.

2) As defined by M. Prather (1994, 1996) and IPCC, the perturbation lifetime of CH₄ is the e-folding time it takes for the CH₄ burden to decay back to its initial value, after a perturbation to methane emissions. For example, if an extra 50 Tg CH₄ is added to the

atmosphere, that perturbation will appear in many later years and decay with an e-fold of perturbation lifetime (from a personal communication with Michael Prather via email. Please also see Holmes et al., 2013 and Nguyen et al., 2020). This study aims to project global CH₄ burdens forced by emission and loss frequency, rather than the response to perturbations.

3) The lifetime used in the box model is derived from transient chemistry-climate simulations, which considered various chemistry-climate factors in determining OH, including the CH₄ feedback. When only the CH₄ emission/concentration is perturbed (like Stevenson et al, 2013 and Heimann et al., 2020), the feedback factor ($f = 1.3$ or similar) is useful to link the changes between CH₄ lifetime and CH₄ abundance, but this is not our case.

Therefore, the turnover time is correct for our calculation of CH₄ burdens. We provide more explanation in the Methods section on the use of CH₄ lifetime (Please see Line 318-323 in the revised manuscript).

Specific comments

L159 NB The influence of temperatures on OH via the temperature-dependent rate coefficient for CH₄+OH will (obviously) be larger in scenarios with more warming, such as SSP370 (which is also discussed in this paper), as compared to SSP126 which is presented here.

Response: Accepted. The global warming effect in the SSP370 scenario is greater on the CH₄ loss fluxes with enhanced rate coefficient. We estimate that the CH₄ lifetimes are shortened by 0.74–1.1 years due to temperature increases alone. Such analysis is added in the revised manuscript. Please see Line 158-164 in the revised manuscript.

L173 Same point as previous one. The impacts of temperature and water vapour will be larger in the SSP370 scenario. These will tend to reduce the longer lifetime related to composition changes. I think this should be more clearly noted.

Response: Accepted. Like Johnson et al. (1999), the impacts of water vapour and temperature changes on CH₄ lifetime is notable in a high forcing scenario (e.g., SSP370). The increase in water vapor enhances the OH primary production by about 25%. We add these analysis and discussions in the revised manuscript. Please see Line 158-164 in the revised manuscript.

L186-188 “Since those models simulate a wide range of OH values and corresponding CH₄

lifetimes (Fig. 3a), the inferred CH₄ emissions will differ in the future.” I think you can (and should) go slightly further with this interpretation. E.g., consider the UKESM results under the SSP126 scenario in Figure S7, which show a CH₄ change of over -50% in 2100. These results were derived using the CMIP6 prescribed CH₄ values (the orange line in Fig. S7), i.e. about -40% in 2100. If UKESM had used emission-driven CH₄ (with SSP126 CH₄ emissions), its CH₄ concentration in 2100 would have been even lower. So the results presented here will underestimate the true changes in methane. This is important to clearly state.

Response: Accepted. We agree with the reviewer’s comment and provide more discussions of the results here. The emission-driven simulation for CH₄ enables a full coupling of CH₄-CO-OH in the atmosphere and allows CH₄ concentrations freely evolves. Given the decreasing CH₄ emissions, the CH₄ concentrations decrease to a larger extent due to the OH feedback. This could not be fully reproduced by the prescribed CH₄ data, as revealed in Figure S7. In other words, the OH enhancement and resulting CH₄ lifetime changes in the emission-driven simulations will be greater than presented here. Also note that in this scenario, the CH₄ feedback is partly masked by the reductions of NO_x and ozone, depending on model structures. Please see these discussions at Line 193-199 in the revised manuscript.

L276-277 “The current CMIP6 simulations are able to reflect the changes in global OH in relation to evolving CH₄ concentrations given in different scenarios.”

Suggest append:

but are not fully self-consistent, as each model simulation infers a different CH₄ trajectory that typically differs from the prescribed CH₄ trajectory for the scenario.

Response: Accepted. We acknowledge this point. The sentence is rewritten as:

The current CMIP6 simulations are able to reflect the changes in global OH in relation to evolving CH₄ concentrations given in different scenarios, but their results are not fully self-consistent, as each model simulation infers its own CH₄ trajectory that typically differs from the prescribed CH₄ trajectory for the scenario.

Please see this sentence at Line 283-286 in the revised manuscript.

L294 Delete “Besides,”

Response: Accepted. We delete this word here. Please see Line 303 in the revised manuscript.

L296 “can represent” -> includes a representation of

Response: Accepted. We reword it to “includes a representation of”. Please see Line 305 in the revised manuscript.

L307 “ τ denotes CH₄ lifetime in years”. I wonder if you should be using the perturbation lifetime here, not the overturning time. See Holmes 2018 (ref 42). This will actually make a rather large difference. Please check.

Response: Accepted. We confirm that the turnover lifetime is appropriate for our calculation here. Holmes 2018 demonstrates the importance of the CH₄ feedback on atmospheric chemistry, which has been discussed in our manuscript (Line 110-113). Please see our detailed explanation in the preceding response.

Reviewer #2 (Remarks on code availability):

Sorry, I haven't checked the code.

Response: We provide a link showing the codes used for analysis, including the instructions and outputs. Please check <https://github.com/mliucn/CMIP6-SSP.git>.

References:

Johnson, C. E., W. J. Collins, D. S. Stevenson, and R. G. Derwent (1999), Relative roles of climate and emissions changes on future tropospheric oxidant concentrations, *J. Geophys. Res.-Atmos*, 104(D15), 18631-18645.

Holmes, C. D., M. J. Prather, O. A. Søvde, G. Myhre (2013). Future methane, hydroxyl, and their uncertainties: key climate and emission parameters for future predictions. *Atmos. Chem. Phys.* 13, 285-302.

Holmes, C. D. (2018), Methane Feedback on Atmospheric Chemistry: Methods, Models, and Mechanisms, *J. Adv. Model. Earth Syst.*, 10(4), 1087-1099, doi:10.1002/2017ms001196.

Heimann, I. et al. (2020) Methane Emissions in a Chemistry-Climate Model: Feedbacks and Climate Response. *J. Adv. Model. Earth Syst.* 12, e2019MS002019.

Nguyen, N. H., Turner, A. J., Yin, Y., Prather, M. J., & Frankenberg, C. (2020). Effects of chemical feedbacks on decadal methane emissions estimates. *Geophysical Research Letters*, 47, e2019GL085706.

Prather, M. J. (1994) Lifetimes and eigenstates in atmospheric chemistry. *Geophysical Research Letters* 21, 801-804.

Prather, M. J. (1996), Time scales in atmospheric chemistry: Theory, GWPs for CH₄ and CO, and runaway growth. *Geophysical Research Letters* 23, 2597-2600.

Prather, M. J., C. D. Holmes, and J. Hsu (2012), Reactive greenhouse gas scenarios: Systematic exploration of uncertainties and the role of atmospheric chemistry, *Geophysical Research Letters*, 39(9), doi:10.1029/2012gl051440.

Stevenson, D. S. et al. (2013) Tropospheric ozone changes, radiative forcing and attribution to emissions in the Atmospheric Chemistry and Climate Model Intercomparison Project (ACCMIP). *Atmos. Chem. Phys.* 13, 3063-3085.

Szopa, S., V. Naik, B. Adhikary, P. Artaxo, T. Berntsen, W.D. Collins, S. Fuzzi, L. Gallardo, A. Kiendler-Scharr, Z. Klimont, H. Liao, N. Unger, and P. Zanis, 2021: Short-Lived Climate Forcers. In *Climate Change 2021: The Physical Science Basis. Contribution of Working Group I to the Sixth Assessment Report of the Intergovernmental Panel on Climate Change* [Masson-Delmotte, V., P. Zhai, A. Pirani, S.L. Connors, C. Péan, S. Berger, N. Caud, Y. Chen, L. Goldfarb, M.I. Gomis, M. Huang, K. Leitzell, E. Lonnoy, J.B.R. Matthews, T.K. Maycock, T. Waterfield, O. Yelekçi, R. Yu, and B. Zhou (eds.)]. Cambridge University Press, Cambridge, United Kingdom and New York, NY, USA, pp. 817–922, doi:10.1017/9781009157896.008

REVIEWERS' COMMENTS

Reviewer #2 (Remarks to the Author):

As noted in earlier reviews, this is an interesting analysis and nicely highlights the importance of the future pathway of OH in determining the evolution of atmospheric CH₄.

The authors have responded to my earlier comments and checked their calculations and updated the manuscript appropriately.

I still have some doubts, but these should not preclude publication of the manuscript in (more or less) its current form. The general ideas presented are sound and robust, and it is for future researchers to check these results with more detailed simulations to see how they stand up to further scrutiny.

Before final publication, I would recommend the authors revise some of the English as it remains a little unclear in a few places (I don't note all these, but, e.g., lines 65-67: "As a comparison, we also analyze the results in the SSP370 scenario, which represents a high-emissions scenario with CO₂ and NTCFs keep increasing." Replace 'with' with 'in which'.) The caption to Supplementary Figure 6 should also note the years/scenario, like the earlier captions.

Reviewer #2 (Remarks to the Author):

“As noted in earlier reviews, this is an interesting analysis and nicely highlights the importance of the future pathway of OH in determining the evolution of atmospheric CH₄.

The authors have responded to my earlier comments and checked their calculations and updated the manuscript appropriately.

I still have some doubts, but these should not preclude publication of the manuscript in (more or less) its current form. The general ideas presented are sound and robust, and it is for future researchers to check these results with more detailed simulations to see how they stand up to further scrutiny.

Before final publication, I would recommend the authors revise some of the English as it remains a little unclear in a few places (I don't note all these, but, e.g., lines 65-67: "As a comparison, we also analyze the results in the SSP370 scenario, which represents a high-emissions scenario with CO₂ and NTCFs keep increasing." Replace 'with' with 'in which'.) The caption to Supplementary Figure 6 should also note the years/scenario, like the earlier captions.”

Response: We check the English writing throughout the manuscript. We revise the following places following the reviewer's suggestions.

Line 65-67: "As a comparison, we also analyze the results in the SSP370 scenario, which represents a high-emissions scenario with continued increases in CO₂ and CH₄."

The caption to Supplementary Figure 6: "These percentage changes were calculated for the Shared Socioeconomic Pathway 1–2.6 scenario during the period of 2015–2100. The purple solid lines denote the tropopause."

-thanks for helping us polish the manuscript.